# Learning to Predict Layout-to-image Conditional Convolutions for Semantic Image Synthesis

**Xihui Liu**
The Chinese University of Hong Kong
xihuiliu@ee.cuhk.edu.hk

**Guojun Yin**
University of Science and Technology of China
gjyin91@gmail.com

**Jing Shao**
SenseTime Research
shaojing@sensetime.com

**Xiaogang Wang**
The Chinese University of Hong Kong
xgwang@ee.cuhk.edu.hk

**Hongsheng Li**
The Chinese University of Hong Kong
hsli@ee.cuhk.edu.hk

## Abstract

Semantic image synthesis aims at generating photorealistic images from semantic layouts. Previous approaches with conditional generative adversarial networks (GAN) show state-of-the-art performance on this task, which either feed the semantic label maps as inputs to the generator, or use them to modulate the activations in normalization layers via affine transformations. We argue that convolutional kernels in the generator should be aware of the distinct semantic labels at different locations when generating images. In order to better exploit the semantic layout for the image generator, we propose to predict convolutional kernels conditioned on the semantic label map to generate the intermediate feature maps from the noise maps and eventually generate the images. Moreover, we propose a feature pyramid semantics-embedding discriminator, which is more effective in enhancing fine details and semantic alignments between the generated images and the input semantic layouts than previous multi-scale discriminators. We achieve state-of-the-art results on both quantitative metrics and subjective evaluation on various semantic segmentation datasets, demonstrating the effectiveness of our approach.[1]

## 1 Introduction

Recently, generative adversarial networks (GAN) [6] have shown stunning results in generating photorealistic images of faces [16, 17] and simple objects [34, 1, 22]. However, generating photorealistic images for complex scenes with different types of objects and stuff remains a challenging problem. We consider semantic image synthesis, which aims at generating photorealistic images conditioned on semantic layouts. It has wide applications on controllable image synthesis and interactive image manipulation. State-of-the-art methods are mostly based on Generative Adversarial Networks (GAN).

A fundamental question to semantic image synthesis is how to exploit the semantic layout information in the generator. Most previous GAN-based approaches feed the label maps as inputs, and generate images by an encoder-decoder network [13, 29, 25]. Nonetheless, since the semantic label maps are only fed into the network once at the input layer, the layout information cannot be well preserved in

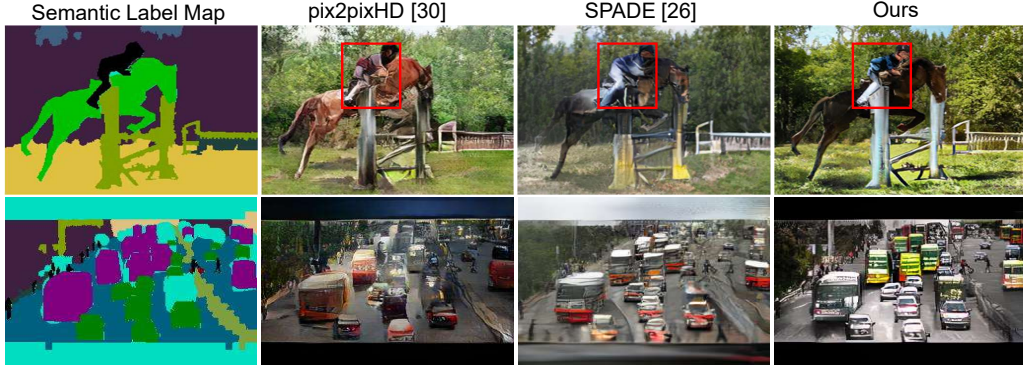

| Semantic Label Map | pix2pixHD [30] | SPADE [26] | Ours |

Figure 1: Semantic image synthesis results by previous approaches and our approach. Best viewed in color. Zoom in for details. Key differences are highlighted by red boxes.

the generator. To mitigate the problem, SPADE [25] uses the label maps to predict spatially-adaptive affine transformations for modulating the activations in normalization layers. However, such feature modulation by simple affine transformations is limited in representational power and flexibility.

On the other hand, we rethink the functionality of convolutional layers for image synthesis. In a generation network, each convolutional layer learns "how to draw" by generating fine features at each location based on a local neighborhood of input features. The same translation-invariant convolutional kernels are applied to all samples and at all spatial locations, irrespective of different semantic labels at different locations, as well as the unique semantic layout of each sample. Our argument is that different convolutional kernels should be used for generating different objects or stuff.

Motivated by the two aforementioned aspects, we propose to predict spatially-varying conditional convolution kernels based on the input semantic layout, so that the layout information can more explicitly and effectively control the image generation process. However, naively predicting all convolutional kernels is infeasible, because it requires a large amount of learnable parameters, which causes overfitting and requires too much GPU memory. Inspired by recent works on lightweight convolutional neural networks [4, 11, 23], we propose to predict the depthwise separable convolution, which factorizes a convolutional operation into a conditional depthwise convolution and a conventional pointwise convolution (*i.e.* $1 \times 1$ convolution). The conditional kernel weights for each spatial location are predicted from the semantic layout by a global-context-aware weight prediction network. Our proposed conditional convolution enables the semantic layout to better control the generation process, without a heavy increase in network parameters and computational cost.

Most existing methods for semantic image synthesis adopt a multi-scale PatchGAN discriminator [29, 25], but its limited representation power cannot match the increased capacity of the generator. We believe that a robust discriminator should focus on two indispensable and complementary aspects of the images: *high-fidelity details*, and *semantic alignment with the input layout map*. Motivated by the two principles, we propose to utilize multi-scale feature pyramids for promoting high-fidelity details such as texture and edges, and exploit patch-based semantic-embeddings to enhance the spatial semantic alignment between the generated images and the input semantic layout.

The contribution of this paper are summarized as follows. (1) We propose a novel approach for semantic image synthesis by learning to predict layout-to-image conditional convolution kernels based on the semantic layout. Such conditional convolution operations enable the semantic layout to adaptively control the generation process based on distinct semantic labels at different locations. (2) We propose a feature pyramid semantics-embedding discriminator which is more effective in encouraging high-fidelity details and semantic alignment with the input layout map. (3) With the proposed approach CC-FPSE, we achieve state-of-the-art results on CityScapes, COCO-Stuff, and ADE20K datasets, demonstrating the effectiveness of our approach in generating images with complex scenes.

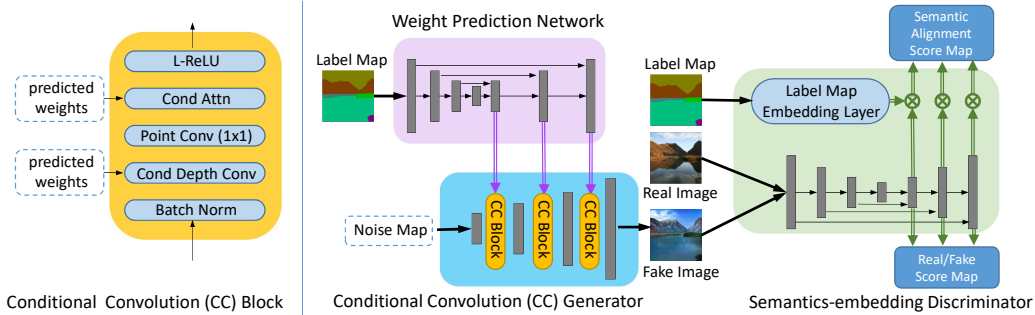

Figure 2: (Left) The structure of a Conditional Convolution Block (CC Block). (Right) The overall framework of our proposed CC-FPSE. The weight prediction network predicts weights for CC Blocks in the generator. The conditional convolution generator is built up of Conditional Convolution (CC) Blocks shown on the left. The feature pyramid semantics-embedding (FPSE) discriminator predicts real/fake scores as well as semantic alignment scores. L-ReLU in the CC Block denotes Leaky ReLU.

## 2   Related Work

**Generative adversarial networks** (GAN) [6] has made great success in image synthesis [1, 17, 22]. Conditional GANs synthesize images based on given conditions, which can be labels [34, 1], sentence descriptions [33, 31], or semantic layout in our task. Our work is also related to image-to-image translation [13, 21], which translates a possible representation of an image into another representation.

**Semantic image synthesis** aims at synthesizing photorealistic images given the semantic layout. Pix2pix [13] adopted an encoder-decoder generator which takes semantic label maps as inputs. Pix2pixHD [29] proposed a coarse-to-fine generator and multi-scale discriminators to generate high-resolution images. SPADE [25] used the semantic label maps to predict affine transformation parameters for modulating the activations in normalization layers. Besides GAN-based approaches, CRN [3] used a cascaded refinement network with regression loss as training supervisions. SIMS [26] developed a semi-parametric approach, by retrieving fragments and refining them with a refinement network. Our method differ from previous GAN-based approaches in how the semantic layout information controls the generation process. We propose to predict spatially-varying convolutional kernels conditioned on the semantic layout, so that it can explicitly control the generation process.

**Dynamic filter networks** [14] was the first attempt to generate dynamic filters based on the inputs. Ha *et al.* [7] proposed HyperNetworks, where a hyper-network is used to generate weights for another network. This idea has been applied to different applications such as neural style transfer [27], super-resolution [15, 12], image segmentation [8, 30], motion prediction [32] and tracking [19]. However, most of them only predicted a limited number of filters, and it would be computation and memory extensive if we use dynamically predicted filters in each layer. Su *et al.* [28] proposed pixel-adaptive CNN which multiplies the conventional convolutional filter with a spatially-varying kernel to obtain convolutional kernels. Zhao *et al.* [35] adopted a shared filter bank and predict adaptive weights to linearly combine the basis filters. Such operations are still based on the conventional convolutions.So the input information has limited capacity in controlling or influencing the adaptive convolutional kernels, and the behaviors of the generation networks were still dominated by the conventional convolutional kernels. Our approach differs from previous work in several aspects. Firstly, we predict the convolutional kernels conditioned on the layout information, so that the conditional information can explicitly control the generation process. Secondly, we reduce the computation and memory costs by introducing depthwise separable convolutions, while enable the conditional information to control the generation process by directly predicting the convolutional kernel weights.

## 3   Method

We propose a novel approach for semantic image synthesis with conditional generative adversarial networks. The proposed framework, CC-FPSE, is composed of a novel generator $G$ with conditional convolutions predicted by the weight prediction network, and a feature-pyramid semantics-embedding

discriminator $D$ as shown in Figure 2 (right). The proposed generator $G$ is able to fully utilize the semantic layout information to control the image generation process by predicting the convolution kernels in multiple layers of the generation network with limited computational resources. The proposed discriminator $D$ is able to supervise the generation of fine details and forces the spatial alignment between the generated image and the input semantic layout by embedding both images and label maps into a joint feature space.

## 3.1 Learning to Predict Conditional Convolutions for Image Generator

Our proposed generator $G$ takes a low-resolution noise map as input. It alternatively uses the proposed conditional convolution blocks [9] and upsampling layers to gradually refine the intermediate feature maps and eventually generate the output image. In conventional convolution layers, the same convolution kernels are applied to all samples and at all spatial locations regardless of their distinct semantic layout. We argue that such convolution operation is not flexible and effective enough for semantic image synthesis. In semantic image synthesis, the convolution layers gradually generate refined features at each location given the coarse features in a local neighborhood. Since different objects or stuff should be generated differently, we would like the convolution layer to be aware of the unique semantic label at the target location.

In order to better incorporate the layout information into the image generation process, we propose to predict convolutional kernel weights based on the semantic layout. Given the input feature map $\mathbf{X} \in \mathbb{R}^{C \times H \times W}$, we aim to produce the output feature map $\mathbf{Y} \in \mathbb{R}^{D \times H \times W}$ by a convolution layer with kernel size $k \times k$. We adopt a weight prediction network that takes the semantic label map as input and outputs the predicted convolutional kernel weights for each conditional convolution layer. However, naively predicting all the kernel weights causes excessive computational costs and GPU memory usage. To solve the problem, we factorize the convolutional layer into depthwise convolution and pointwise convolution, and only predict the weights of the lightweight depthwise convolutions.

### 3.1.1 Efficient Conditional Convolution Blocks for Image Generation

A conventional convolution kernel has $D \times C \times k \times k$ weight parameters. A naive solution for generating the spatially-varying convolution kernel needs to predict $D \times C \times k \times k \times H \times W$ weight parameters. This is impractical because the convolution operation is the basic building blocks of the generator $G$ and would be stacked for multiple times in the generator. Such a network is not only computation and memory intensive, but also prone to overfit the training data.

To solve the problem, we introduce depthwise separable convolution [4] and only predict the depthwise convolutional kernel weights, which substantially reduces the number of parameters to predict. In particular, we factorize the convolutional kernel into a conditional depthwise convolution and a conventional pointwise convolution (*i.e.*, $1 \times 1$ convolution). The conditional depthwise convolution performs spatial filtering over each input channel independently, and its spatially-varying kernel weights are dynamically predicted based on the semantic layout. The predicted weights for the conditional convolution layer are denoted as $\mathbf{V} \in \mathbb{R}^{C \times k \times k \times H \times W}$, and the output feature maps are denoted as $\mathbf{Y} \in \mathbb{R}^{C \times H \times W}$. The conditional depthwise convolution is formulated as,

$$\mathbf{Y}_{c,i,j} = \sum_{m=0}^{k-1} \sum_{n=0}^{k-1} \mathbf{X}_{c,i+m,j+n} \mathbf{V}_{c,m,n,i,j}, \tag{1}$$

where $i, j$ denotes the spatial coordinates of the feature maps, $k$ denotes the convolution kernel size, and $c$ denotes the channel index. The $C \times H \times W$ convolutional kernels in $\mathbf{V}$ with kernel size $k \times k$ operates at each channel and each spatial location of $\mathbf{X}$ independently to generate output feature maps. Then we exploit a conventional pointwise convolution ($1 \times 1$ convolution) to map the $C$ input channels to $D$ output channels, and the output is denoted as $\mathbf{Y}' \in \mathbb{R}^{D \times H \times W}$.

In addition, we also propose a conditional attention operation to gate the information flow passed to the next layer. The conditional attention weights are predicted in the same way as the conditional convolution kernels, which will be detailed later. An element-wise product between the predicted attention weights $\mathbf{A} \in \mathbb{R}^{C \times H \times W}$ and the convolution output $\mathbf{Y}'$ produces the gated feature maps,

$$\mathbf{Z}_{c,i,j} = \mathbf{Y}'_{c,i,j} \mathbf{A}_{c,i,j}, \tag{2}$$

where $c$ is the channel index and $i, j$ denotes the spatial location in the feature maps.

The size of predicted parameters in the conditional convolution and the conditional attention are $C \times k \times k \times H \times W$ ($k = 3$ in our implementation) and $C \times H \times W$, respectively. The parameter size is reduced by $D$ times compared to directly predicting the whole convolutional kernel weights.

By predicting unique convolutional kernel weights for each spatial location, the image generation process becomes more flexible and adaptive to the semantic layout conditions. In the meantime, we keep an affordable parameter size and computational cost by introducing the depthwise separable convolutions. We define a ResBlock-like structure, named Conditional Convolution Block, with the operations introduced above. As shown in Figure 2 (left), it includes a conventional batch normalization layer, a conditional depthwise convolution with $k = 3$, a conventional pointwise convolution, followed by a conditional attention layer, and finally the the non-linear activation layer. There are also identity additive skip connections for evert two such blocks.

### 3.1.2 Conditional Weight Prediction and Overall Generator Structure

The conditional weight prediction network predicts the conditional weights $\mathbf{V}$ given the input semantic layout. A simple design of the weight prediction network would be simply stacking multiple convolutional layers. In SPADE [25], two convolutional layers of kernel size $3 \times 3$ are applied to the downsampled semantic label map to generate the adaptive scale and bias for their proposed adaptive normalization layer. But downsampling a semantic label map to a very small size, *e.g.*, $8 \times 8$, by nearest neighbor interpolation will inevitably lose much useful information. Moreover, such a structure only has a receptive field of $5 \times 5$, which restricts the weight prediction from incorporating long-range context information. If there is a large area of the same semantic label, pixels inside this area can only access a $5 \times 5$ local neighborhood with identical semantic labels. So they will be processed by identical predicted weights, regardless of their relative positions inside the object or stuff.

Therefore, we design a global-context-aware weight prediction network with a feature pyramid structure [20]. The architecture of our weight prediction network is shown in Figure 2 (right). The label map is first downsampled through the layout encoder, and then upsampled by the decoder with lateral connections from the encoder. The features at different levels of the feature pyramid are concatenated with the original semantic map to obtain the global-context-aware semantic feature maps, which are used to predict the conditional convolution weights and conditional attention weights separately. We use two convolutional layers to predict the conditional convolution weights. To predict the conditional attention weights, we adopt two convolutional layers and a Sigmoid activation layer.

With the encoder-decoder structure of the weight prediction network, our predicted weights are aware of not only the local neighborhood, but also long-range context and relative locations.

The overall generator network $G$ is built of a series of Conditional Convolution Blocks and upsampling layers, with conditional weights predicted by the weight prediction network .

## 3.2 Feature Pyramid Semantics-embedding Discriminator

We believe that a good discriminator should focus on two indispensable and complementary aspects: *high-fidelity details such as texture and edges*, and *semantic alignment with the input semantic map*. Existing methods for semantic image synthesis apply a multi-scale PatchGAN discriminator [29, 25], where images concatenated with the semantic label maps are scaled to multiple resolutions and fed into different discriminators with identical structure. But it still struggles to discriminate the fine details, and does not pose strong constraints on the spatial semantic alignment between the generated image and the input label map.

Motivated by the aforementioned two design principles of discriminators, we propose a more effective design for the discriminator $D$. We create multi-scale feature pyramids for promoting high-fidelity details such as texture and edges and exploit a semantics-embedding discriminator to force the spatial semantic alignment between the generated images and the input semantic layout.

### 3.2.1 Feature Pyramid Discriminator

Current image generation methods tend to generate images with blurry edges, textures and obvious artifacts. This problem suggests that we should cast more attention on low-level details when designing the discriminator architectures. On the other hand, the discriminator should also have a global view

of the high-level semantics. The previously introduced multi-scale PatchGAN discriminator [29] attempts to balance large receptive field and fine details by multiple discriminators at different scales. The same image at different scales are independently fed into different discriminators, leading to increased network parameters, memory footprint and computational cost.

Inspired by the evolution from image pyramids to feature pyramids [20], we propose a single feature pyramid discriminator to produce a multi-scale feature representation with both global semantics and low-level texture and edge information. As shown in Figure. 2(right), our feature pyramid discriminator takes the input image at a single scale. The bottom-up pathway produces a feature hierarchy consisting of multi-scale feature maps and the top-down pathway gradually upsamples the spatially coarse but semantically rich feature maps. The lateral combines the high-level semantic feature maps from the top-down pathway with the low-level feature maps from the bottom-up pathway. As a result, the combined multi-scale features are semantically strong, as well as containing finer low-level details such as edges and textures. So the discriminator would pose stronger constraints on both the semantic information and the fine details.

### 3.2.2 Semantic Embeddings for Discriminator

In the conventional discriminators for semantic image synthesis, an image and its corresponding semantic label map is concatenated and fed into the discriminator as its inputs. However, there is no guarantee that the discriminator makes use of the label maps for distinguishing real/fake images. In other words, the discriminator could satisfy the training constraints by only discriminating whether an image is real or not, without considering whether it matches well with the label map. Inspired by projection discriminator [24] which computes the dot product between the class label and image feature vector as part of the output discriminator score, we adapt this idea to our scenerio where the condition is the spatial label map. In order to encourage the semantic alignment between generated images and the conditional semantic layout, we propose a patch-based semantics embedding discriminator.

Our discriminator takes only the real or generated images as inputs, and produces a set of feature pyramids $\{\mathbf{F}_1, \mathbf{F}_2, \mathbf{F}_3\}$ at different scales. $\mathbf{F}_i \in \mathbb{R}^{C \times N_i \times N_i} (i \in \{1, 2, 3\})$ denotes feature maps at a spatial resolution of $N_i \times N_i$ with $C$ channels. The feature vector at each spatial location of $\mathbf{F}_i$ represents a patch in the original image. The conventional PatchGAN discriminator tries to classify if each patch is real or not, by predicting a score for each spatial location in the feature map $\mathbf{F}_i$. While we force the discriminator to classify not only real or fake images, but also whether the patch features match with the semantic labels in that patch within a joint embedding space.

We downsample the label map to the same spatial resolution as $\mathbf{F}_i$, and embed the one-hot label at each spatial location into a $C$-dimensional vector. The embedded semantic layout is denoted as $\mathbf{S}_i \in \mathbb{R}^{C \times N_i \times N_i}$. We calculate the inner product between each spatial location of $\mathbf{F}_i$ and $\mathbf{S}_i$, to obtain a semantic matching score map, where each value represents the semantic alignment score of the corresponding patch in the original image. The semantic matching score is added with the conventional real/fake score as the final discriminator score. In this way, not only does the discriminator guide the generator to generate high-fidelity images, but also it drives the generated images to be better semantically aligned with the conditional semantic layout.

### 3.3 Loss Functions and Training Scheme

The generator and the discriminator of our network are trained alternatively, where the discriminator adopts the hinge loss for distinguishing real/fake images while the generator is optimized with multiple losses, including the hinge-based adversarial loss, discriminator feature matching loss, and perceptual loss, following previous works [29, 25],

$$L_D = -\mathbb{E}_{(x,y)}[min(0, -1 + D(x,y))] - \mathbb{E}_{z,y}[min(0, -1 - D(G(z,y),y))], \tag{3}$$

$$L_G = -\mathbb{E}_{(z,y)}D(G(z,y),y) + \lambda_P \mathbb{E}_{(z,y)}L_P(G(z,y),x) + \lambda_{FM}\mathbb{E}_{(z,y)}L_{FM}(G(z,y),x), \tag{4}$$

where $x$ is a real image, $y$ is the semantic label map, and $z$ is the input noise map. $L_P(G(z,y),x)$ denotes the perceptual loss, which matches the VGG extracted features between the generated images and the original images. $L_{FM}(G(z,y),x)$ denotes the discriminator feature matching loss, which matches the discriminator intermidiate features between the generated images and the original images. $\lambda_P$ and $\lambda_{FM}$ denote the weights for the perceptual loss and feature matching loss, respectively.

# 4 Experiments

## 4.1 Datasets and Evaluation Metrics

We experiment on Cityscapes [5], COCO-Stuff [2], and ADE20K [36] datasets. The Cityscapes dataset has 3,000 training images and 500 validation images of urban street scenes. COCO-Stuff is the most challenging dataset, containing 118,000 training images and 5,000 validation images from complex scenes. ADE20K dataset provides 20,000 training images and 2,000 validation images from both outdoor and indoor scenes. All images are annotated with semantic segmentation masks.

We evaluate our approach from three aspects. We firstly compare synthesized images by our approach and previous approaches, and conduct a human perceptual evaluation to compare the visual quality of the generated images. We then evaluate the segmentation performance of the generated images using a segmentation model pretrained on the original datasets. We use the same segmentation models as those in [25] for testing. The segmentation performance is measured by mean Intersection-over-Union (mIOU) and pixel accuracy. Finally, we calculate the distribution distances between the generated images and real images by the Fréchet Inception Distance (FID) [10].

## 4.2 Implementation Details

The training and generated image resolution is $256 \times 256$ for COCO-Stuff and ADE20K datasets, and $256 \times 512$ for Cityscapes dataset. For the generator, synchronized batch normalization between different GPUs is adopted for better estimating the batch statistics. For the discriminator, we utilize instance normalization. We use Leaky ReLU activations, to avoid sparse gradients caused by ReLU activation. We adopt ADAM [18] optimizer with learning rate $0.0001$ for the generator and $0.0004$ for the discriminator. The weights for the perceptual loss $\lambda_P$ is 10 and the weight discriminator feature matching loss $\lambda_{FM}$ is 20. Following [25], to enable multi-modal synthesis and style-guided synthesis, we apply a style encoder and a KL-Divergence loss with loss weight 0.05. Our models are trained on 16 TITANX GPUs, with a batch size of 32. We train 200 epochs for Cityscapes and ADE20K datasets, and 100 epochs for COCO-Stuff dataset. Code is available at https://github.com/xh-liu/CC-FPSE.

## 4.3 Qualitative Results and Human Perceptual Evaluation

We compare our results with previous approaches pix2pixHD [29] and SPADE [25], as shown in Figure 3. The images generated by our approach show significant improvement over previous approaches for challenging scenes. They have finer details such as edges and textures, and less artifacts, and matches better with the input semantic layout. Figure 4 shows more images generated by our proposed approach. More results and comparisons are provided in the supplementary material.

We also conduct a human perception evaluation to compare the generated image quality between our method and the previous state-of-the-art method, SPADE [25]. In particular, we randomly sample 500 semantic label maps from the validation set of each dataset. At each experiment, the worker is shown a semantic label map with two generated images by our approach and SPADE, respectively. The worker is required to choose an image with higher quality that matches better with the semantic layout. We found that in Cityscapes, COCO-Stuff, and ADE20K datasets respectively, 55%, 76%, and 61% images generated by our method is preferred compared to SPADE. The human perceptual evaluation validates that our approach is able to generate higher-fidelity images that are better spatially aligned with the semantic layout.

## 4.4 Quantitative Results

Table 1 shows the segmentation performance and FID scores of results by our approach and those by previous approaches. CRN [3] uses cascaded refinement networks with regression loss, without using GAN for training. SIMS is a semi-parametric approach which retrieves reference segments from a memory bank and refines the canvas by a refinement network. Both pix2pixHD [29] and SPADE [25] are GAN-based approaches. Pix2pixHD takes the semantic label map as the generator input, and uses a multi-scale generator and multi-scale discriminator to generate high-resolution images. SPADE takes a noise vector as input, and the semantic label map are used for modulating the activations in normalization layers by learned affine transformations. Our approach performs consistently better than previous approaches, which demonstrate the effectiveness of the propose approach. Note that

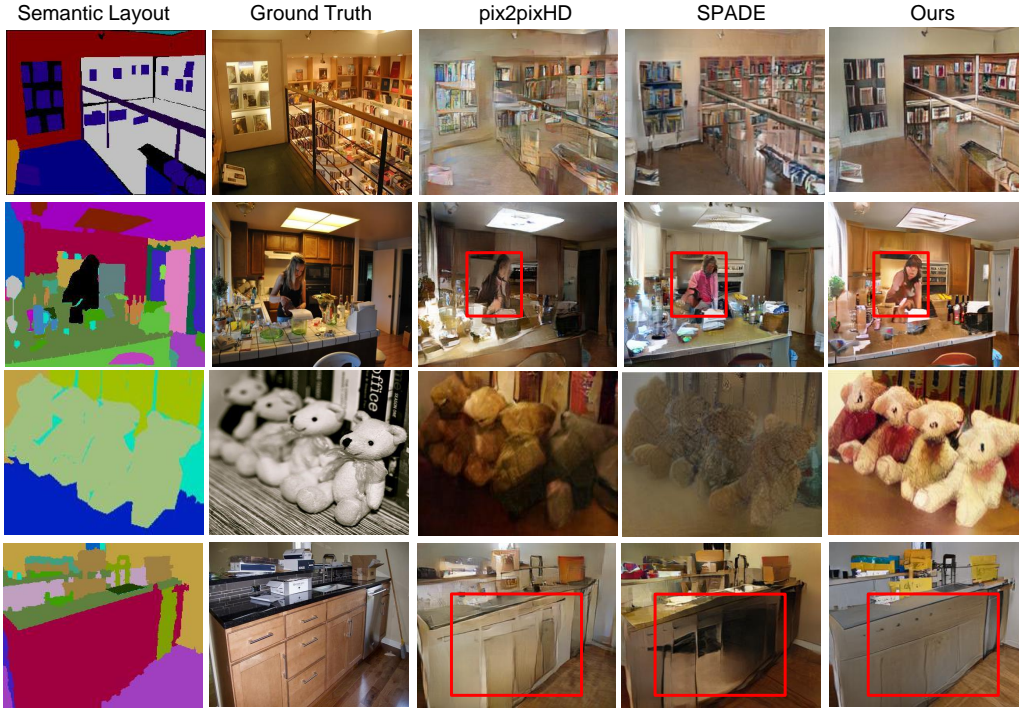

Figure 3: Results comparison with previous approaches. Better viewed in color. Zoom in for details.

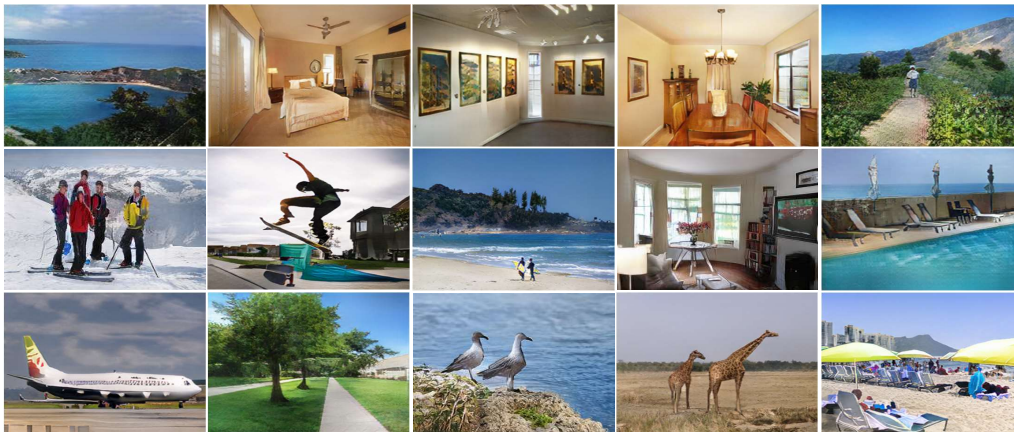

Figure 4: Semantic image synthesis results on COCO and ADE20K. Better viewed in color.

SIMS has better FID scores than GAN-based approaches, because it generates images by refining segments retrieved from the real data. However, it has poor segmentation performance, because it might retrieve semantically mismatched patches.

## 4.5 Ablation Studies

We conduct controlled experiments to verify the effectiveness of each component in our approach. We use the SPADE [25] model as our baseline, and gradually add or eliminate each component to the framework. Our model is denoted as CC-FPSE in the last column. The segmentation mIOU scores of the generated images by each experiment are shown in Table 2.[2]

Table 1: Results by the proposed and previous approaches on multiple public datasets. Higher mIOU/accuracy and lower FID score indicate better performance.

| | COCO-Stuff | | | Cityscapes | | | ADE20K | | |
|---|---|---|---|---|---|---|---|---|---|
| | mIOU ↑ | Accu ↑ | FID ↓ | mIOU ↑ | Accu ↑ | FID ↓ | mIOU ↑ | Accu ↑ | FID ↓ |
| CRN [3] | 23.7 | 40.4 | 70.4 | 52.4 | 77.1 | 104.7 | 22.4 | 68.8 | 73.3 |
| SIMS [26] | N/A | N/A | N/A | 47.2 | 75.5 | **49.7** | N/A | N/A | N/A |
| pix2pixHD [29] | 14.6 | 45.7 | 111.5 | 58.3 | 81.4 | 95.0 | 20.3 | 69.2 | 81.8 |
| SPADE [25] | 37.4 | 67.9 | 22.6 | 62.3 | 81.9 | 71.8 | 38.5 | 79.9 | 33.9 |
| Ours | **41.6** | **70.7** | **19.2** | **65.5** | **82.3** | 54.3 | **43.7** | **82.9** | **31.7** |

Table 2: Ablation studies on COCO-Stuff dataset.

| | Baseline | (1) | (2) | (3) | (4) | (5) | (6) | CC-FPSE (Ours) |
|---|---|---|---|---|---|---|---|---|
| Generator | SPADE | CC w/o FP | CC w/ FP | CC w/o FP | SPADE w/ FP | SPADE w/ FP | CC w/ FP | CC w/ FP |
| Discriminator | MsPatch | MsPatch | MsPatch | FP+SE | MsPatch+SE | FP+SE | MsPatch+SE | FP+SE |
| mIOU | 35.2 | 36.2 | 36.7 | 40.4 | 38.0 | 39.17 | 40.4 | 41.3 |

**Conditional convolutions for generator.** We firstly replace the SPADE layer with our conditional convolution layers to incorporate the semantic layout information in the experiments denoted as "CC". By comparing baseline with (1) (CC generator vs SPADE generator, both with MsPatch discriminator), (5) with CC-FPSE (ours) (CC generator vs SPADE generator, both with FPSE discriminator), and (4) with (6) (CC generator vs SPADE generator, both with MsPatch+SE discriminator), results indicate that our conditional convolutions are able to better exploit the semantic layout information for adaptively generating high-quality images.

**Feature pyramid weight prediction network.** Next, we replace the feature pyramid structure with a stack of two convolutional layers in the weight prediction network, and this experiment is denoted as "w/ FP" and "w/o FP". Comparing (1) with (2), and (3) with CC-FPSE (Ours), we found that removing the feature pyramid structure for the weight prediction network leads to inferior performance, indicating that the global and long-range information are necessary for predicting the convolutional weights.

**FPSE Discriminator.** We fix our proposed generator ("CC w/ FP" or "SPADE w/ FP") and test different designs of the discriminator, to demonstrate the effectiveness of our FPSE discriminator. We force the spatial semantic alignment with the semantic layout, by the introduced semantics-embedding constraint for the discriminator. Comparing (2) with (6) indicates the effectiveness of the semantics embedding discriminator. With the semantics-embedding constraint, the discriminator is driven to classify the correspondence between the image patches and the semantic layout. So the generator is encouraged to generate images that are better aligned with the semantic layout. Furthermore, we replace the multiscale discriminator with the feature pyramid structure, denoted as "FP+SE", which is our proposed discriminator design. The comparison between (6) and last column CC-FPSE (Ours) indicates that the feature pyramid discriminator structure, which combines the low-level and semantic features at different scales, leads to further performance improvement.

## 5 Conclusion

We propose a novel approach (CC-FPSE) for image synthesis from a given semantic layout via better using the semantic layout information to generate images with high-quality details and well aligned semantic meanings. Our generator is able to better exploit the semantic layout to control the generation process, by predicting the spatially-varying weights for the conditional convolution layers. Our feature pyramid semantics-embedding discriminator guides the generator to generate images that contain high-fidelity details and aligns well with the conditional semantic layout. Our approach achieves state-of-the art performance and is able to generate photorealistic images on Cityscapes, COCO-Stuff, and ADE20K datasets.

**Acknowledgments**

This work is supported in part by SenseTime Group Limited, in part by the General Research Fund through the Research Grants Council of Hong Kong under Grants CUHK14202217, CUHK14203118, CUHK14207319, CUHK14208417, CUHK14239816, and in part by CUHK Direct Grant. We thank Lu Sheng for proofreading and helpful suggestions on the paper.

## Footnotes

[1]Code is available at https://github.com/xh-liu/CC-FPSE

[2]To be comparable with ablation study results in [25], we report the model performance at 50 epochs.

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
