[Supplementary Material]

# Learning to Predict Layout-to-image Conditional Convolutions for Semantic Image Synthesis Supplementary Materails

**Xihui Liu**
The Chinese University of Hong Kong
xihuiliu@ee.cuhk.edu.hk

**Guojun Yin**
University of Science and Technology of China
gjyin91@gmail.com

**Jing Shao**
SenseTime Research
shaojing@sensetime.com

**Xiaogang Wang**
The Chinese University of Hong Kong
xgwang@ee.cuhk.edu.hk

**Hongsheng Li**
The Chinese University of Hong Kong
hsli@ee.cuhk.edu.hk

Examples of generated images by our approach from COCO-Stuff, Cityscapes, and ADE20K datasets respectively, are shown in Figure 1. Over proposed approach is able to synthesis images of diverse scenes. Moreover, we show the semantic image synthesis results compared to previous approaches pix2pixHD and SPADE in Figure 2. Some differences between the generated images of different approaches are highlighted in red boxes. Our proposed approach generates high-quality images with fine details. It can generate small objects based on the label map, while previous approaches are likely to ignore them. For example, in the first row of Figure 2, our approach generates a driver inside the bus based on the semantic layout, while other approaches fails to generate the driver.

Figure 1: Semantic image synthesis results by our proposed approach. Images are generated from label maps in the validation set of COCO-Stuff dataset, Cityscapes dataset, and ADE20K dataset, respectively. Best viewed in color. Zoom in for details.

| Semantic Layout | pix2pixHD | SPADE | Ours |
|---|---|---|---|

Figure 2: Semantic image synthesis results by previous approaches and our approach. Best viewed in color. Zoom in for details.