[Reviews · NeurIPS 2019]

Reviewer 1



The idea to predict the parameters of the convolutions is interesting, however I cannot really understand the motivation behind the proposed method. If I understand correctly, the V layer is only smaller than a full convolutional layer by a factor of D, (which is 3?). I see the real bottleneck in the fact that you need to predict H*W kernels although you know the semantic content at each pixel. To me it would make much more sense to predict DxCxKxKxL channels, where L is the number of labels. if you would use the conditional weight prediction, you would still be able to use context as input. With respect to the discriminators it is difficult to understand what the authors developed and what has been taken from other papers. Although they reference other papers, they do not state, what they have contributed. It is really hard to judge performance. The number of parameters must be high, especially when including attention as well. Especially since the network details are not listed anywhere (number of parameters etc), it is hard to compare to SPARSE. So many different adjustments are made that it is hard to determine which one is the most important one. The Ablation study is going in the right direction but is lacking some parts. For example, CondConv pred C/O FP + MsPatch. Only when you do not use the feature pyramid and no attention, you can actually assess whether the parameter learning is better than SPADE. Also, it would be good to see results with SPADE + FP and SPADE + FP+SE. Originality: medium Quality: medium Clarity: medium - without having read SPADE, the reader does not know what how to compare these approaches. Significance: low-medium

Reviewer 2



This paper proposes a strongly conditional network for generating images from semantic maps. How impacted is this network by small changes in the input map - for example given 3 sequential frames of a video (as segmentation maps) - is the model consistent in assigning colors and structures? Or do small changes in the geometry of the semantic objects have a large impact on the output? This is mostly curiousity, as having smoothness inherent in the model has large potential for video applications. Some amount of qualitative results comparing to other models were shown, but showing the important regions of the input conditioning, and the influence of input perturbations on the model output could also lead to valuable insight - using something like GradCAM or related methods may be possible for checking the importance of input features. In 4.3 (qualitative worker analysis) there could be more detail (variance across labelers / uncertainty / statistical significance) rather than a pure percentage preference. How many workers labeled the 500 images? Given that this is largely a (very impressive) empirical paper, it would be nice to see a larger exploration of ablation on various components, or some larger intuition on how and why the network was designed how it was. The empirical results are convincing, and the demonstrated experiments are thorough - though more ablations can add greater insight, the current experiments seem sufficient given the high quality of the model. I strongly encourage the authors to release their code, as the community should be able to use, improve, and extend this work in interesting new ways - perhaps doing some "in-the-wild" ablation studies along the way. Feedback post-rebuttal: My score remains unchanged primarily because I had no major criticisms of this paper to begin with - the response didn't fundamentally change my perception of the work. The author's comments clarified some of my key questions, thank you for the explanations.

Reviewer 3



Originality: This paper mainly proposes two things: 1. Generate weights of CNNs from the input semantic label for the generator. 2. Use feature pyramids throughout the network (i.e., in both generator and discriminator) I think this combination is new for the semantic image synthesis task. Related studies are well cited and the paper is different enough from them. Quality: The method is reasonable and the idea to use separable convolutions to reduce the number of parameters makes sense. The proposed method is compared with previous methods and better results are reported. One thing is that the author claims that their method is better than SPADE, but the comparison is done for the original implementation of SPADE but not for the idea. I think that it should be possible to estimate the scaling parameters instead of the CNN weights using the same network. It would be valuable to perform such experiments. Clarity: The paper is easy to follow and understand. I personally prefer to see more explanations on each loss term. It is not always very clear what is actually used by knowing the name of the loss (e.g. perceptual or FM). Significance: The proposed method is reasonable given the current state of the arts and the combination seems novel. I expect that this will attract researchers working on similar fields.

[Author Response · NeurIPS 2019]

Table 1: More ablation studies.

| | Cityscapes | | | | | COCO-Stuff | | |
|---|---|---|---|---|---|---|---|---|
| Generator | SPADE | SPADE | SPADE | SPADE w/ FP | CondGen w/FP | SPADE | SPADE w/ FP | CondGen w/FP |
| Discriminator | MsPatch | FP | FP+SE | FP+SE | FP+SE | MsPatch | FP+SE | FP+SE |
| mIOU | 62.3 (baseline) | 62.9 (**R1**) | 63.11 (**R1**) | 64.17 (**R1&R3**) | 64.9 (ours) | 37.4 (baseline) | 38.79 (**R1&R3**) | 40.1 (ours) |

Figure 1: A sequence of generated images from Cityscapes.

**To reviewer #1**

**Q1: (1) Motivation behind the proposed method? (2) V layer is only smaller than a convolution layer by a factor $D$ (which is 3?). (3) Make more sense to predict $D \times C \times K \times K \times L$ kernels.**

A1: (1) The key idea is to generate spatially-varying convolution kernels at different spatial locations according to the input layouts, so that the image synthesis process can be better controlled by the semantic layouts. (2) V layer is smaller by a factor of $D$. However, $D$ is the number of input channels and generally ranges from 64 to 1024 in different layers. (3) The reviewer's suggestion on predicting $D \times C \times K \times K \times L$ kernels has two limitations: (3.1) it takes more parameters than our design. If we predict $D \times C \times K \times K \times L$ kernels, the last layer of the weight prediction network would be a fully-connected layer with $D \times C \times K \times K \times L$ output dimensions. While in our design, the last layer of the weight prediction network is a convolution layer with $C \times K \times K$ output dimensions. For instance, $L = 182$ for COCO dataset, $K = 3$, and $C$ and $D$ ranges from 64 to 1024.(3.2) The kernels are fixed for all spatial locations with the same label, regardless of their distinct contexts. For instance, a pixel in the top and bottom regions of a "person" would share convolutional kernels, which cannot effectively achieve appearance variations for pixels with the same label.

**Q2: Contribition of the discriminator design.**

A2: The contribution of our discriminator design is twofold. (1) This is the first time to adopt multi-scale feature pyramids in the discriminator to promote high-fidelity details such as textures and edges. (2) The patch-based semantic embeddings are adopted to enhance the spatial semantic alignment between the generated images and input semantic layout. The idea is inspired by projection discriminator which computes the dot product between the class label and image feature vector, but we are the first to adapt this idea to the spatial label map and make it work on semantic image synthesis. We will add more explanations in the final version.

**Q3: Number of parameters and more ablation studies comparing with SPADE.**

A3: The parameter size of our proposed generater is 107.4 million, which is similar to that of SPADE (96 million). Although predicting convolutional kernels requires more parameters than predicting scale and bias vectors in SPADE, we predict only 1/3 of the convolutional layer weights (as shown in CC Block in original Fig. 2), while SPADE predicts all BN parameters. Following the reviewer's suggestions, we did more ablation studies as shown in Table 1.

**To Reviewer #2**

**Q4: How impacted is this network by small changes in the input map?**

A4: We conduct an experiment on generating videos. We extract segmentation maps from a video sequence in Cityscapes by a semantic segmentation model, and use those segmentation maps and the same noise vector to generate a sequence of images. As shown in Figure 1, the generated results for adjacent frames are smooth and consistent.

**Q5: Human perceptual evaluation details.**

A5: Each pair of images is annotated by 5 workers independently, and in total 20 workers involved in the human evaluation. The 5 workers reach an agreement for 70% of the cases.

**Q6: More ablation studies and code.**

A6: More ablation studies are shown in Table 1. We will release code upon paper acceptance.

**To Reviewer #3**

**Q7: Compare the SPADE-like method (learned scaling parameters) to the proposed method (predicting conditional convolution) with the same discriminator and weight prediction network (SPADE w/ FP + FP+SE).**

A7: The comparisons on COCO and Cityscapes are shown in Table 1, which demonstrates the effectiveness of predicting conditional convolution (ours) over predicting scaling parameters for BN (SPADE).

[Meta-Review · NeurIPS 2019]

All reviewers are in unanimous agreement for acceptance. The paper has a number of interesting contributions, mostly empirical, in utilizing a conditional weights network and feature pyramids. As promised in your rebuttal, please release code before acceptance.